# Experimental and Theoretical Insights into the Synergistic Effect of Iodide Ions and 1-Acetyl-3-Thiosemicarbazide on the Corrosion Protection of C1018 Carbon Steel in 1 M HCl

**DOI:** 10.3390/ma13215013

**Published:** 2020-11-06

**Authors:** Aeshah Hassan Alamri

**Affiliations:** Chemistry Department, College of Science, Imam Abdulrahman Bin Faisal University, P.O. Box 76971, Dammam 31441, Saudi Arabia; ahalamri@iau.edu.sa

**Keywords:** 1-acetyl-3-thiosemicarbazide, acid corrosion, C1018 carbon steel, electrochemistry, potassium iodide

## Abstract

Experimental insights into the synergistic effect of 1-acetyl-3-thiosemicarbazide (AST) and iodide ions on the corrosion of C1018 carbon steel in 1 M HCl solution were investigated using open-circuit potential (OCP), linear polarization resistance (LPR), electrochemical frequency modulation (EFM), potentiodynamic polarization (PDP) measurements and electrochemical impedance spectroscopy (EIS). Theoretical studies were further undertaken using ACD/LABS Percepta software, density functional theory (DFT) calculations and Monte Carlo simulation to understand the mechanism of the corrosion inhibition process and interpret the experimental results at the atomic and molecular levels. The electrochemical results obtained showed that AST alone inhibited the acid-induced corrosion of C1018 carbon steel. The inhibition efficiency increases with a concentration reaching up to 72.27% at 750 ppm of AST. The addition of 5 mM KI to 250 ppm of AST improved the inhibition efficiency to 81.64%. The solubility and protonated state results predicted using the ACD/LABS Percepta software showed that AST was highly soluble in the aqueous acidic medium and approximately 95% of AST exists in the neutral form in 1 M HCl (pH = 0). DFT calculations and a Monte Carlo simulation were utilized to predict the active reactivity sites of AST and calculate the lowest adsorption energy and configuration of AST alone and AST + iodide on/Fe (110)/water interface.

## 1. Introduction

Carbon steel, such as C1018, are widely used in industries as construction materials for engineering applications due to their low cost and ease of fabrication. However, carbon steel has low resistance to acid corrosion [1]. Several efforts were recently made to develop technologies to mitigate the adverse effect of acid corrosion of carbon steel. These methods include protective coatings, cathodic protection, corrosion-resistant materials and corrosion inhibitors. The most practical and economical way to control and prevent aqueous corrosion is the use of corrosion inhibitors [2,3]. Corrosion inhibitors work by surface adsorption onto the corroding metal [4].

Organic compounds successfully applied as corrosion inhibitors are compounds possessing heteroatoms such as N, O, P, S and cyclic rings in their molecules. In this regard, nitrogen-containing structures such as benzimidazole [5,6], imidazole [7,8], tetrazole [9,10] and triazole [11,12] have been extensively studied as carbon steel corrosion inhibitors in acid media. Compounds containing nitrogen and sulfur atoms in the same structures have been shown to possess even higher corrosion inhibition efficiency for carbon steel than those containing only nitrogen. In this regard, we can cite some compounds, such as benzothiazoles [13,14] and thioureas [15,16]. Apart from benzothiazoles and thioureas, other chemical compounds possessing these structural characteristics are thiosemicarbazide and its derivatives.

Thiosemicarbazides and its derivatives have found several applications in the medical field, including antifungal, antitumor, antiviral antibacterial and antimalarial therapeutics [17]. Investigations into the corrosion inhibition potential of thiosemicarbazide and its derivatives in acid media have been reported in recent years [18,19,20,21,22]. However, most studies indicate that the inhibition efficiencies obtained were rather low and require improvements [23,24].

Synergism has been identified as an acceptable way to further improve the corrosion protection of several organic corrosion inhibitors [25,26]. The use of iodide ions as a synergist in enhancing corrosion inhibitors performances has been extensively documented [27,28]. However, information on the synergistic effect between thiosemicarbazides and halide ions and other synergists is lacking in the literature.

In order to further improve the performance of thiosemicarbazide as a corrosion inhibitor, we investigated the corrosion inhibition efficiency of a new thiosemicarbazide, AST, alone and in the presence of potassium iodide for C1018 carbon steel corrosion in 1 M HCl. Electrochemical techniques such as open-circuit potential (OCP), linear polarization resistance (LPR), electrochemical frequency modulation (EFM), potentiodynamic polarization (PDP) and electrochemical impedance spectroscopy (EIS) were adopted in this study. Theoretical studies were undertaken using ACD/LABS Percepta software, DFT calculations and a Monte Carlo simulation to elucidate atomistic information on the corrosion inhibition action of AST on the C1018 surface.

## 2. Experimental Section

### 2.1. Material and Sample Preparation

The carbon steel sample used in this study was a cylindrical sample of grade UNS G10180 (C1018) with the following chemical composition (wt.%): C = 0.16, Mn = 0.7, S = 0.05, P = 0.04 and Fe balance. The C1018 samples had a dimension of 3/8” in diameter and 1/2” in length with a surface area of 5.23 cm^2^. Before beginning the experiment, the C1018 samples were abraded from 400 to 1000 grit silicon carbide papers. They were subsequently degreased with acetone, rinsed with deionized water, dried using air and kept in a desiccator prior to use in electrochemical measurements. 1-Acetyl-3-thiosemicarbazide (AST) and potassium iodide (KI) were purchased from Sigma-Aldrich. The corrosive medium, i.e., 1 M HCl, was prepared using deionized water and analytical reagent grade 37% HCl. The AST concentrations used for this study were 250, 500, 750 and 1000 ppm, while a fixed concentration of 5.0 mM KI was used to investigate the effect of iodide on the performance of AST.

### 2.2. Electrochemical Measurements

The Gamry Reference 1010E Potentiostat/Galvanostat (Gamry Instruments, Warminster, PA, USA) and Echem Analyst software suite (Gamry Instruments, Warminster, PA, USA) for data analysis were utilized for the electrochemical measurements. The jacketed version was used for the Gamry EuroCell electrochemical glass cell kit. For this experiment, three-port arrangements, consisting of the cylindrical C1018 carbon steel as the working electrode, the saturated calomel electrode (SCE) as the reference electrode and a cylindrical graphite rod as the counter electrode, were employed. Electrochemical techniques such as LPR, EFM, PDP and EIS were adopted in this study. Each of the electrochemical experiments began after the C1018 carbon steel was immersed in the 1 M HCl for 3600 s (1 h) to attain a steady-state OCP.

The EIS measurements frequency was 100 kHz–0.1 Hz with an AC amplitude of 10 mV. LPR was performed within ±20 mV/E_corr_ using 0.167 mV/s as the scan rate. For EFM, two frequencies, 2 and 5 Hz, were used. The base frequency used was 1 Hz and 16 cycles were measured, respectively. Finally, the PDP curves were measured with potentials from −250 to +250 mV vs. SCE using a 0.5 mV/s scan rate. All the electrochemical experiments were conducted at 25 °C in a static unstirred condition. The electrochemical corrosion measurements were carried out by immersing the C1018 into the 1 M HCl solutions in which different concentrations of AST and AST + KI were added.

### 2.3. ACD/LABS Predictions of Solubility and Protonation of AST

Solubility and the protonation of AST in 1 M HCl (pH = 0) were predicted, respectively, using the ACD/LABS Percepta 14.3.0 (Build 3044) software (Advanced Chemistry Development, Inc. Toronto, ON, Canada). The protonation prediction and speciation of AST were carried out in pH range from 0–14.

### 2.4. Density Functional Theory (DFT) Calculations

Geometry optimization of AST molecules in neutral and protonated forms was performed using DFT calculations. The well-known generalized gradient approximation (GGA) via Becke-Lee-Yang-Parr (BLYP) functional combined with the double numerical polarization (DNP) basis set (GGA/BLYP/DNP) level of theory was used. Fermi smearing of 0.005 Ha and a 3.5 real spatial cutoff was used to improve computational performance. The solvent (water) effect was included using the COSMO setting with a dielectric constant of 78.54. All the parameters were set to fine to ensure accurate results. The HOMO and LUMO orbitals and Fukui function plots of AST in neutral and protonated forms were obtained and discussed. The DMol^3^ module in the BIOVIA Materials Studio software version 8 (Dassault Systemes, Waltham, MA, USA) was adopted for the DFT calculations.

### 2.5. Monte Carlo Simulations

The lowest adsorption energy and configuration of the neutral form of AST, which is the most abundant form of the molecule at pH = 0 (1 M HCl), was used in the Monte Carlo simulation, using the ADSORPTION LOCATOR module in the Materials Studio software (version 8.0, BIOVIA). The simulation was carried out based on the theory of Metropolis Monte Carlo [29,30]. The simulation involved cleaving bulk Fe into the Fe (110) surface and building a supercell of 10 × 10 with a vacuum thickness of 30 Å. ATS was optimized using the COMPASS force field and added with 400 water molecules into the supercell. The total energy and adsorption energy for the Fe (110) configuration, the most stable inhibitor, were obtained after the simulation.

## 3. Results and Discussion

### 3.1. Solubility and Protonation Analysis of AST

Corrosion inhibitors must be soluble in aqueous solution in order to migrate to the metal surface and inhibit metal corrosion. In this regard, the solubility of AST was predicted using the ACD/LABS Percepta 14.3.0 (Build 3044) software. To determine the nature of the AST molecules in 1 M HCl (pH = 0), the protonation prediction and speciation of AST were carried out at a pH range from 0 to 14. Figure 1a shows the molecular structure, while Figure 1b shows the solubility vs. the pH profile of 1-acetyl-3-thiosemicarbazide (AST) predicted using the software. The results obtained from Figure 1 indicate that the solubility of AST was 63,300 mg/L at pH = 0. This indicates that AST is highly soluble in 1 M HCl and can serve as a good corrosion inhibitor. Figure 2 shows the protonation process of AST molecules in acidic solution. According to the figure, AST with a pka value of 9.8 is highly likely to be protonated in acidic solution. To further understand the different forms of AST present in an aqueous phase in a pH range from 0 to 14, a plot of percentage abundance of the microspecies against pH and three different forms of AST are shown in Figure 3a,b, respectively. The results presented in Figure 3 indicate that at pH = 0, approximately 95% of AST exists in the neutral form corresponding to state PS2 and 0.5% in the protonated form corresponding to state PS1.

Before any electrochemical experiment is conducted, the working electrode must attain a stable OCP with time in the corrosive solution with or without a corrosion inhibitor. Figure 4 shows the OCP with time for C1018 carbon steel in 1 M HCl (blank) and with the addition of 250, 500 and 750 ppm of AST, respectively. The results from Figure 4 indicate that the carbon steel reached a stable OCP in 1 M HCl (blank) and with different concentrations of AST after 1 h immersion. The final OCP for the blank solution was more negative but became positive with the addition of different concentrations of AST. This is evidence that AST was able to inhibit the corrosion of carbon steel [31]. As a rule, OCP is stable when the potential fluctuates around <3 mV [32]. The results obtained from Figure 4 show that the OCP was stable; therefore, other electrochemical measurements were undertaken at the OCP.

### 3.2. Linear Polarization Resistance (LPR) Measurements 

In the LPR technique, a small variation in potential usually less than ±20 mV is applied around the corrosion potential, and the current response is measured. A plot of current versus potential is made, which gives a slope known as polarization resistance (*R_p_*). The *R_p_* can be related to the corrosion current density (*i_corr_*), according to the Stern–Geary equation [33]. The *R_p_* obtained is also inversely proportional to the corrosion rate (CR). Electrochemical kinetic parameters were obtained from LPR measurements for C1018 carbon steel in 1 M HCl (blank), and the various concentrations of AST and AST + KI at 25 °C are presented in Table 1. The inhibition efficiency (*η* (%)) was obtained according to Equation (1):(1)$$\eta_{LPR}\left(\%\right)=\frac{R_{p}^{inh}-R_{p\;}}{R_{p\;}}\times 100$$
where *R_p_^inh^* is the polarization resistance in the presence of the inhibitor and *R_p_* is the polarization resistance in the absence of the inhibitors (blank solution).

The *R_p_* values increase in the presence of the corrosion inhibitor compared to the blank. For instance, the *R_p_* value at 750 ppm of AST was 59.47 ohms compared with 16.69 ohms in the uninhibited solution. The RP values were seen to increase further with the addition of 5 mM KI to AST. The *R_p_* values increased to 89.82 ohms with 250 ppm AST + 5 mM KI. However, *R_p_* values were observed to decrease when the concentration of AST was increased beyond 250 ppm, even in the presence of 5 mM KI. The increase in the inhibition efficiency with the concentration of AST could be attributed to the increased adsorption of AST molecules on the carbon steel surface. The presence of 5 mM KI did not improve the inhibition efficiency of AST, especially at higher concentrations. The presence of 750 ppm of AST gave the highest inhibition efficiency of 72.27%. The addition of 5 mM KI to 750 ppm of AST gave an inhibition efficiency of 77.69%, which did not significantly improve the inhibition efficiency. However, a lower concentration of AST (250 ppm) + 5 mM KI gave the highest, and best, inhibition efficiency of 81.64%. Beyond this concentration, the adsorption of the molecules reaches a saturation point and AST molecules begin to desorb from the carbon steel surface even with an increase in AST concentration. This subsequently leads to a decrease in inhibition efficiency, as observed in Table 1. No reasons or conclusions for this behavior were given in a similar study [34]. This phenomenon requires further investigations using an advanced sensitive surface analytical instrument, such as X-ray photoelectron spectroscopy (XPS), to quantify the amount of AST and KI on the surface of the C1018 steel. 

### 3.3. Potentiodynamic Polarization (PDP) Measurements

Potentiodynamic polarization (Tafel) plots were obtained to investigate further the anodic and cathodic electrochemical processes that occur on the C1018 carbon steel surface in 1 M HCl (blank) and various concentrations of AST and AST + KI at 25 °C. Table 2 presents the various kinetic electrochemical corrosion parameters, such as the corrosion current density (*i_corr_*), corrosion potential (*E_corr_*), corrosion rates (*mpy*) and anodic and cathodic Tafel slopes (*β_a_, β_c_*) obtained from the extrapolation of the anodic and cathodic Tafel lines. The inhibition efficiency (*η_PDP_ (*%)) was obtained from the corrosion current density in 1 M HCl (*i_corr_*) and in the presence of AST ($i_{corr}^{inh}$) according to Equation (2):(2)$$\eta_{PDP}\left(\%\right)=\frac{i_{corr}-i_{corr}^{inh}}{i_{corr}}\;\times \;100$$

From the Tafel results presented in Table 2, it is clear that an increase in the concentration of AST leads to an increase in the inhibition efficiency and a decrease in the corrosion current density (*i_corr_*), suggesting the formation of a more stable AST film on the carbon steel surface as the concentration increases. This film impedes the flow of corrosive ions (H^+^) to the metal surface and decreases the corrosion rate. The presence of KI is seen to further decrease the corrosion current density of the carbon steel corrosion in 1 M HCl, especially at a concentration of 250 ppm + 5 mM KI. Beyond this concentration, there was a sharp increase in *i_corr_*.

Figure 5 shows the potentiodynamic polarization (Tafel) curves for C1018 carbon steel in 1 M HCl and different concentrations of AST and AST + KI. It could be observed from the figure that the presence of the corrosion inhibitor additives did not have any effect on the *E_corr_*. The *E_corr_* did not shift to the anodic side or cathodic side, showing that AST acts as a mixed-type corrosion inhibitor. This behavior is also seen in the presence of AST + KI. The values of the Tafel slopes (*β_a_, β_c_*) obtained from the extrapolation of the anodic and cathodic Tafel lines did not change much with the addition of AST to the corrosive medium. This suggests that AST protects the carbon steel by blocking the reaction sites of the carbon steel surface without any modification of the corrosion mechanism. Similar behavior has been reported by other organic molecules [28,35]. In all cases, the highest inhibition efficiency obtained was 79.30% at 250 ppm + 5 mM KI, which agrees with the LPR results.

### 3.4. Electrochemical Frequency Modulation (EFM) Measurements

The application of EFM in corrosion inhibition research has recently been reviewed [36]. The conclusion is that EFM is an efficient and excellent electrochemical technique for the investigation of corrosion inhibitors, especially in acidic media. The fact that EFM can determine the value of *i_corr_* directly without prior knowledge of the Tafel constants makes it very useful for electrochemical corrosion investigation. The electrochemical kinetic parameters, such as the *i_corr_* and Tafel constants (*β_a_*) and (*β_c_*), were obtained directly from EFM results, as presented in Equations (3)–(5). The activation model was used to obtain the results since the corrosion reaction takes place in the active mode (1 M HCl). The performance or the accuracy of EFM can be judged based on the so-called causality factors (CF2 and CF3), as presented in Equations (6) and (7).
(3)$$i_{corr}=\frac{{\left(i_{\omega_{1}.\omega_{2}}\right)}^{2}}{2\sqrt{8i_{\omega_{1}.\omega_{2}}i_{2\omega_{2}\pm \omega_{1}}-3{\left(i_{\omega_{1}.\omega_{2}}\right)}^{2}\;}}$$
(4)$$\beta_{c}=\frac{U(i_{\omega_{1}.\omega_{2}})}{(-i_{\omega_{2}\pm \omega_{1}})+\sqrt{8i_{\omega_{1}.\omega_{2}}i_{2\omega_{1}\pm \;\omega_{2}}-3{\left(i_{\omega_{1}.\omega_{2}}\right)}^{2}\;}}$$
(5)$$\beta_{a}=\frac{U(i_{\omega_{1}.\omega_{2}})}{(i_{\omega_{2}\pm \omega_{1}})+\sqrt{8i_{\omega_{1}.\omega_{2}}i_{2\omega_{2}\pm \;\omega_{1}}-3{\left(i_{\omega_{1}.\omega_{2}}\right)}^{2}\;}}$$
(6)$$CF2=\frac{i_{\omega_{1}\pm \omega_{2}}}{i_{2\omega_{1}}}=2$$
(7)$$CF3=\frac{i_{2\omega_{1}\pm \omega_{2}}}{i_{3\omega_{1}}}=3$$

The inhibition efficiency (*η_EFM_ (*%)) from electrochemical frequency modulation (EFM) was obtained from (*i_corr_*) in 1 M HCl and (*i_corr_^inh^*) in the presence of AST according to Equation (8):(8)$$\eta_{EFM}\left(\%\right)=\frac{i_{corr}-i_{corr}^{inh}}{i_{corr}}\;\times \;100$$

Table 3 presents the EFM data for C1018 carbon steel in 1 M HCl and various concentrations of AST and AST + KI at 25 °C. The *i_corr_* value was 701.30 µA·cm^−2^ without AST (1 M HCl). The addition of various concentrations of AST further reduces the corrosion current density. For example, the addition of 750 ppm of AST cut the *i_corr_* value by almost one-half to 350.40 µA·cm^−2^. This corresponds to an inhibition efficiency of 50.04%. The addition of KI to AST is similarly seen to further reduce the corrosion current density compared to the 1 M HCl solution. For instance, the addition of 250 ppm AST + 5 mM KI reduced the *i_corr_* value to 254.70 µA·cm^−2^, corresponding to an inhibition efficiency of 63.69%. This is an indication of the synergistic effect between the iodide ions and AST. A further increase in AST concentrations with a fixed amount of KI (5 mM) led to increased *i_corr_* values. The *β_c_* values are slightly higher than the *β_a_* values in all cases, i.e., in the presence of AST and AST + KI, suggesting that the corrosion reaction is slightly mechanistically cathodic [37]. The EFM results presented also show the closeness of CF2 and CF3 values to the theoretical values of two and three, verifying the assumptions of the EFM theory.

### 3.5. Electrochemical Impedance Spectroscopy (EIS) Measurements

EIS measurements were further used to obtain the corrosion characteristics of the C1018 carbon steel and the electrochemical parameters in the absence and presence of AST and in the presence of AST + KI at 25 °C. Important mechanistic information on the corrosion kinetics was obtained from the EIS measurements. Figure 6a,b shows the Nyquist and Bode form of the EIS results for C1018 carbon steel in 1 M HCl and different concentrations of AST and AST + KI, respectively.

As shown in the Nyquist curves, the addition of AST to 1 M HCl increased the diameters of the spectra. The Nyquist curves are further increased after the introduction of AST + KI. This shows that the addition of AST into the corrosive acidic medium alone and in the presence of KI reduces the corrosion rate of C1018 carbon steel and protects it against corrosion. All the Nyquist curves deviated from perfect semicircles, and the shapes are similar in all cases, which indicates that the addition of AST and AST + KI to the acid solution does not affect the mechanism of corrosion of C1018 carbon steel. Such deviations from ideal semicircles were attributed to frequency dispersions and surface inhomogeneity of the carbon steel [38]. The shape shows a corrosion process due to charged transfer, as reported elsewhere [39].

The Bode plots presented in Figure 6b show an increase in the absolute values with AST compared with 1 M HCl. These values increase further with AST + KI, indicating better protection of C1018 carbon steel. The values are seen to drop after 250 ppm AST + 5 mM KI in agreement with other electrochemical methods.

Figure 7a presents the electrical circuit model elements used to fit the EIS data for C1018 carbon steel in 1 M HCl solution, AST and AST + KI. Figure 7b shows the corresponding fitting plot using the blank (1 M HCl, 250 ppm AST and 250 ppm AST + KI). Equation (9) was used to calculate the double-layer capacitance (*C_dl_*) and Equation (10) was used in calculating inhibition efficiency, as earlier reported [1].
(9)$$C_{dl}=Y_{o}{\left(2\pi f_{\max}\right)}^{n–1}\;$$(10)$$\eta_{EIS}\left(\%\right)=\frac{R_{ct\;}^{inh}-\;R_{ct}}{R_{ct}}\;\times \;100$$

As observed from Table 4, the values of *R_ct_* increase, whereas those of *C_dl_* decrease in the presence of AST alone and AST + KI. This shows that the additives adsorbed on the C1018 carbon steel surface, protecting the metal against corrosion by the acid solution. It also shows that there is a decrease in the local dielectric constant and an increase in the double-layer thickness at the steel–solution interface.

### 3.6. Comparison of the Inhibition Efficiency Obtained by Different Electrochemistry

Figure 8 presents a comparison of the inhibition efficiencies obtained from the different electrochemical techniques: EFM, LPR, PDP and EIS at 250 ppm AST and 250 ppm AST + 5 mM KI. The results show that all the electrochemical techniques employed in this study were in agreement with each other, although there was variation in the values of the inhibition efficiencies obtained. The inhibition efficiencies obtained from LPR, PDP and EIS measurements were in very close agreement, and only the results obtained from EFM differ more pronouncedly. This may be due to the underlying assumptions of the EFM techniques [36]. Table 5 presents a comparison of the performance of the proposed corrosion inhibitor with similar ones already reported in the literature. It is evident that AST showed improved corrosion inhibition performance as compared to others already reported.

### 3.7. Density Functional Theory (DFT) Calculations

DFT was further utilized to evaluate the active sites responsible for the adsorption of AST onto the C1018 carbon steel. Based on the analysis of the protonation state of AST presented in Section 3.1, it was predicted that at pH = 0, approximately 95% of AST exists in the neutral form and 0.5% in the protonated form. The DFT calculations were essential to understand the relationship between the molecular structure of AST and its ability to inhibit C1018 corrosion in 1 M HCl. Figure 9a presents the optimized geometric structures, Figure 9b presents molecular orbitals and Figure 9c presents the Fukui function for AST and ASTH. It is clear from Figure 9 that the HOMO and the LUMO orbitals are seen to spread around the entire molecules in both AST (neutral) form and ASTH (protonated form). However, more of the electrons were visible on the S4 atom. This indicates that the S atom is likely to be the main active site along with the contribution from N1, N2, N5 and O7 atoms. These active sites are responsible for donating electron density to the metal surface and accepting electrons from the metal surface during the interaction of the molecule with the C1018 steel’s surface.

More information about the local reactivity of AST on the steel surface can be gleaned using the Fukui function [40,41]. The electrophilic Fukui function (*f*^+^) shows the region in a molecule more likely to accept electrons from a chemical species, whereas the nucleophilic Fukui function (*f*^−^) indicates the sites likely to donate electrons to a chemical species. Figure 9c indicates that the Fukui function shows a similar trend as that of the HOMO and LUMO molecular orbitals. It is evident that the high values of *f*^+^ are localized more at S4 and O7 atoms of AST in both neutral and protonated forms. In the same vein, the highest values of *f*^−^ values were found in S4 and O7 atoms for AST (neutral form) and S4 and C3 in the AST (protonated form).

### 3.8. Monte Carlo Simulations

Monte Carlo simulation has recently emerged as one of the preferred theoretical tools by several researchers in understanding the adsorption of corrosion inhibitors with metal surfaces [42,43,44]. In this study, the neutral form of AST was selected to simulate the interaction between the corrosion inhibitor and Fe surface. A similar simulation was conducted to investigate the enhanced efficiency of AST with the addition of iodide ions. Figure 10a,b shows the snapshots for the adsorption of AST and AST + I^−^, respectively, on Fe (110)/400 H_2_O using the Monte Carlo simulation. It is evident that the AST molecules adopt almost a flat configuration on the Fe (110) surface due to the distribution of electron density across all the heteroatoms present on AST. Such an orientation ensures a strong interaction between AST and the Fe surface. A similar configuration is also evident with AST + iodide.

The adsorption energy for the interaction between corrosion inhibitors and metal surfaces is a measure of the stability of the inhibitor–metal interface. More negative values of the adsorption energy indicate stronger adsorption of corrosion inhibitor molecules on the metal surface and, by extension, potential corrosion inhibition [45]. Figure 11 shows a plot of the total energy and adsorption energy for the AST and AST + I^−^ systems on the Fe (110)/400 H_2_O interface at 25°C. It is clear from the results that AST + iodide has both the highest negative total energy and adsorption energy than AST alone with the Fe (110)/400 H_2_O Interface. This indicates that the presence of iodide has a synergistic effect and enhances the adsorption energy of AST with the Fe surface. Corrosion inhibition synergism between AST and iodide is due to ion-pair interactions between the corrosion inhibitor and iodide ion, which increase the surface coverage on the metal surface [46].

## 4. Conclusions

1-Acetyl-3-thiosemicarbazide (AST) showed the promising corrosion protection of C1018 carbon steel in 1 M HCl. The protection was enhanced in the presence of KI.Inhibition efficiency increases with an increase in the concentration of AST. The highest inhibition efficiency of 81.69% was obtained with 250 ppm AST + 5 mM KI. All the electrochemical results were in good agreement.PDP results showed that the presence of AST alone and with AST + KI could simultaneously affect both the cathodic and anodic electrochemical corrosion reaction i.e., mixed-typed corrosion inhibitor.DFT calculations and a Monte Carlo simulation reveal that sulfur, nitrogen and oxygen atoms present in AST were the reactivity sites. The AST molecule had a strong interaction with the Fe surface. The interaction was further enhanced in the presence of iodide.The electrochemical studies were in good agreement with the results obtained from DFT calculations and Monte Carlo simulations.

## Figures and Tables

**Figure 1 materials-13-05013-f001:**
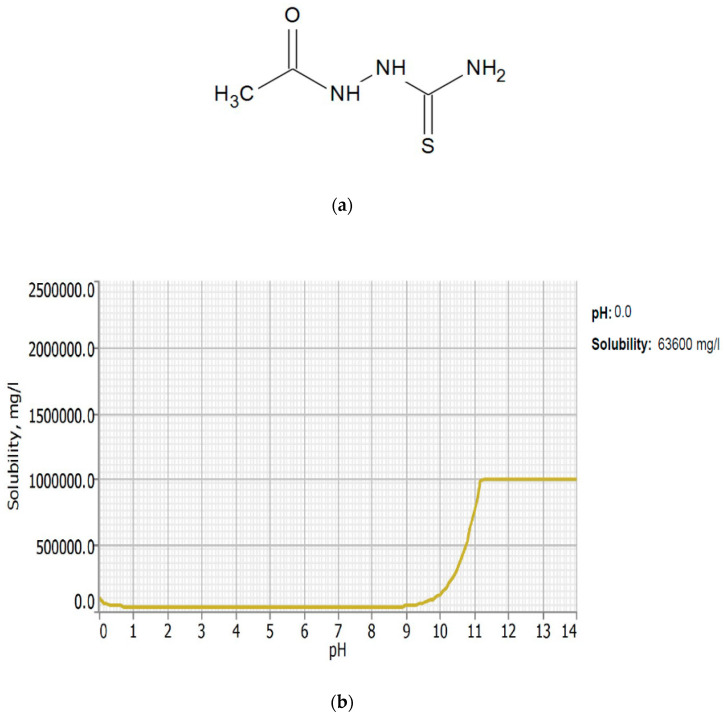
Molecular structure (**a**) and the solubility vs. pH (**b**) of 1-acetyl-3-thiosemicarbazide (AST).

**Figure 2 materials-13-05013-f002:**
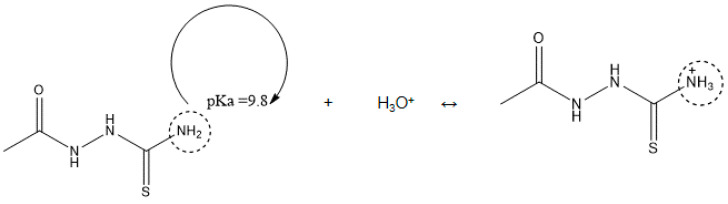
The protonation process of AST molecules in acidic solution.

**Figure 3 materials-13-05013-f003:**
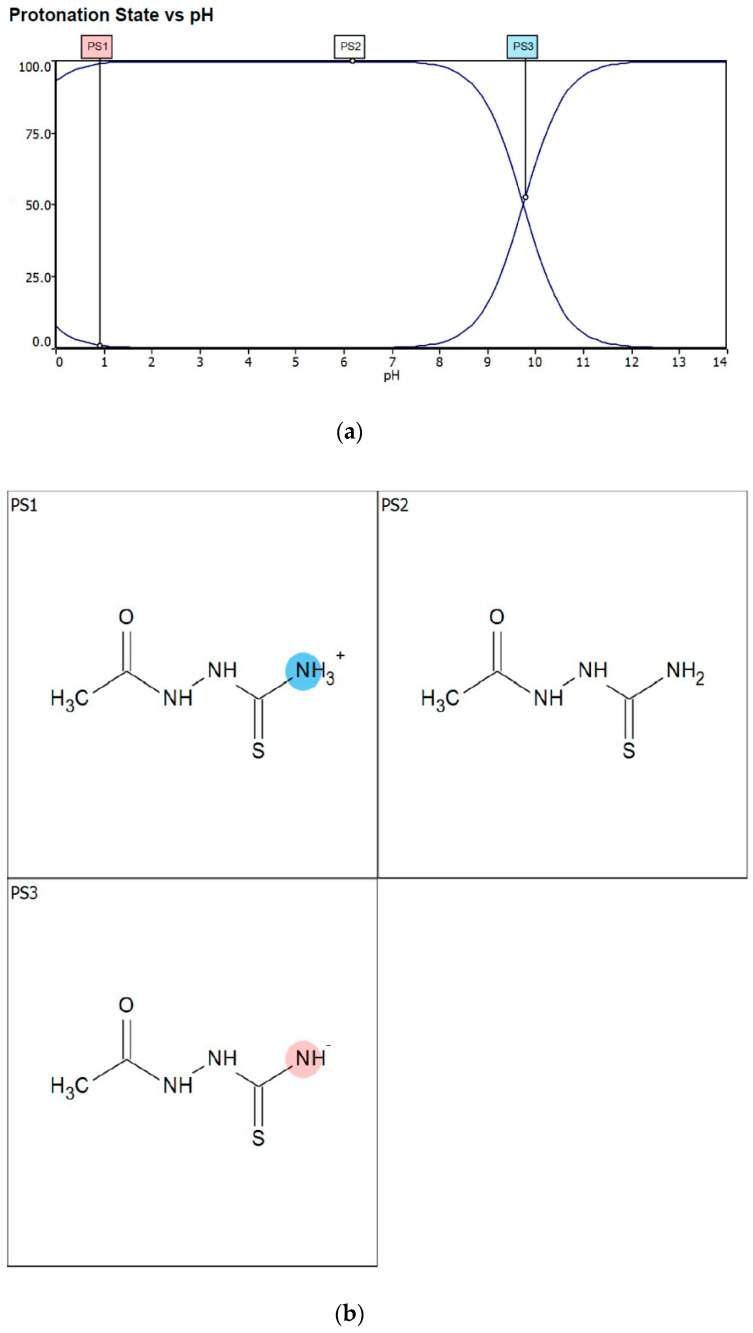
(**a**) The protonation process of AST molecules at different pH values and (**b**) the percentage of the three different microspecies at different pH values. At pH = 0, approximately 95% of AST exists in state PS2 and 0.5% in state PS1.

**Figure 4 materials-13-05013-f004:**
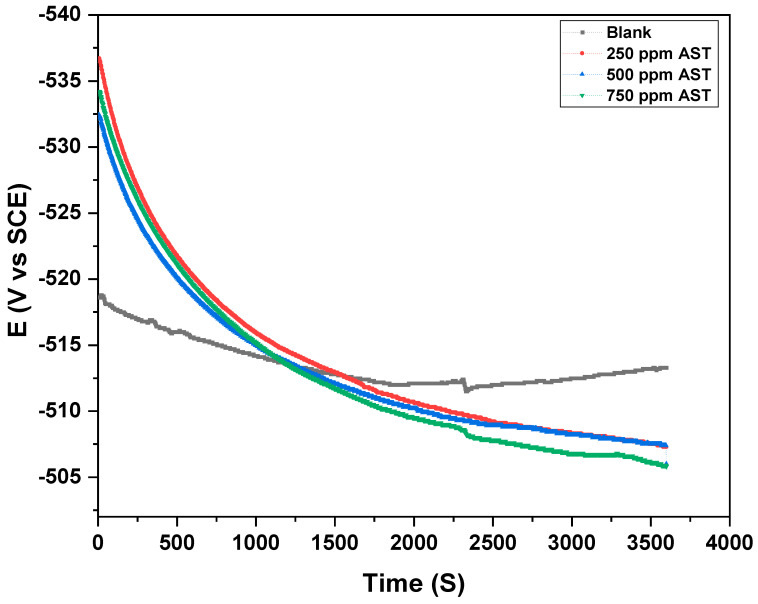
OCP for C1018 carbon steel in 1 M HCl with various concentrations of AST inhibitors.

**Figure 5 materials-13-05013-f005:**
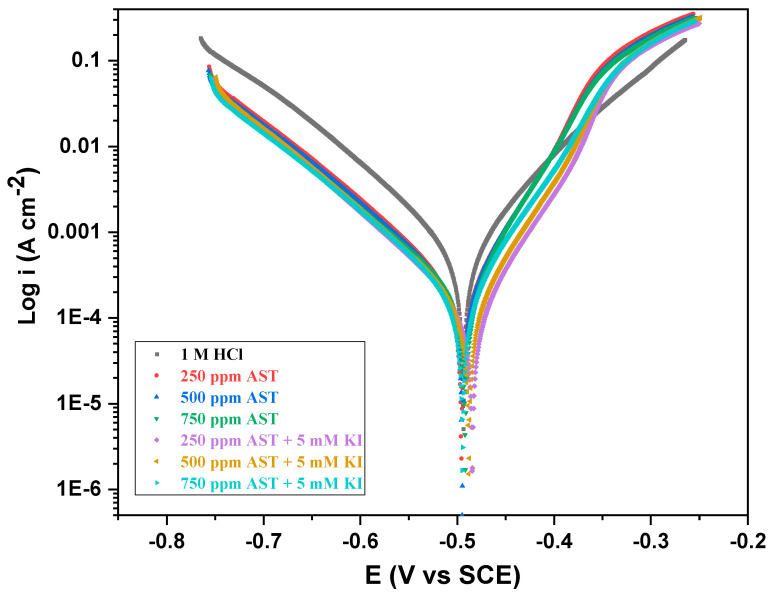
Potentiodynamic polarization (Tafel) curves for C1018 carbon steel corrosion in 1 M HCl with various concentrations of AST inhibitors and AST + KI.

**Figure 6 materials-13-05013-f006:**
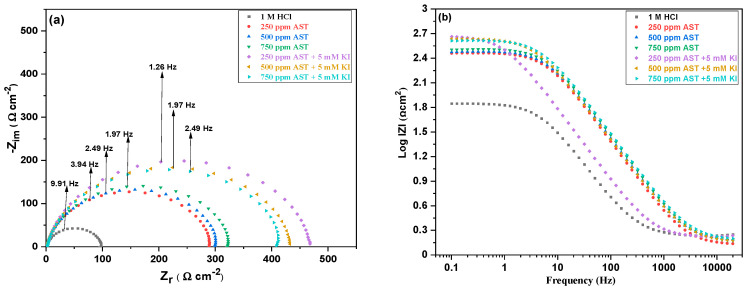
(**a**) Nyquist and (**b**) Bode plots for C1018 carbon steel in 1 M HCl with various concentrations of AST inhibitors and AST + KI.

**Figure 7 materials-13-05013-f007:**
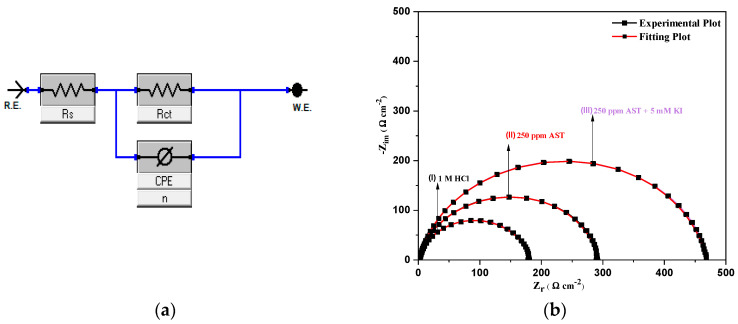
(**a**) Equivalent circuit model used to fit the EIS data for C1018 carbon steel in 1 M HCl solution and (**b**) corresponding fitting plots.

**Figure 8 materials-13-05013-f008:**
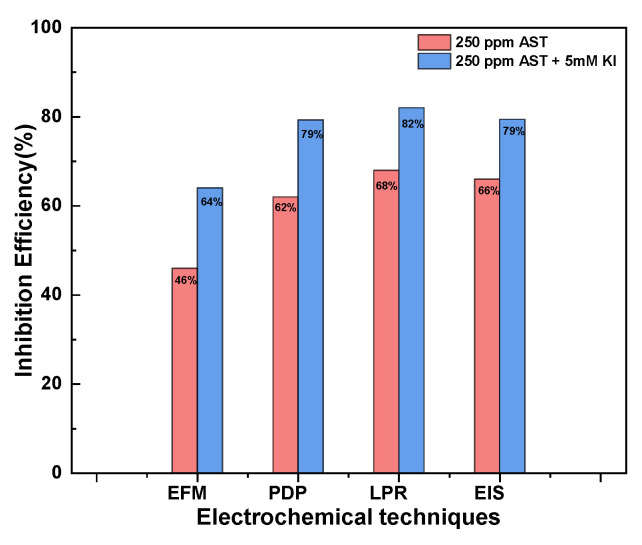
Comparison of the Inhibition efficiencies obtained from the different electrochemical techniques: EFM, LPR, PDP and EIS with 250 ppm AST and 250 ppm AST + 5 mM KI.

**Figure 9 materials-13-05013-f009:**
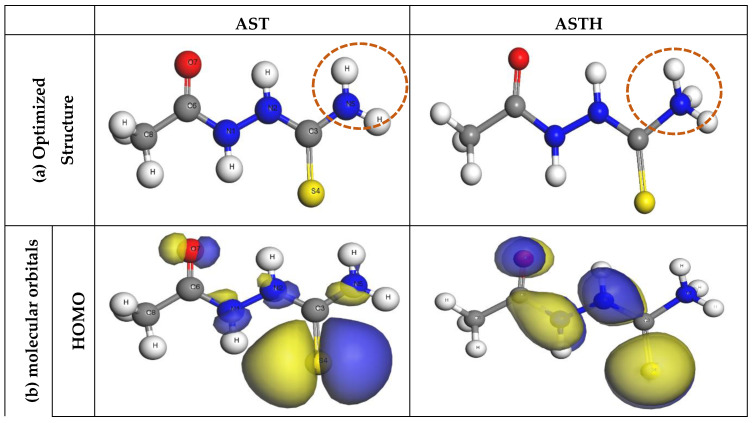
(**a**) Optimized geometric structures, (**b**) molecular orbitals and (**c**) Fukui function for AST and ASTH at the GGA/BLYP/DNP level of theory.

**Figure 10 materials-13-05013-f010:**
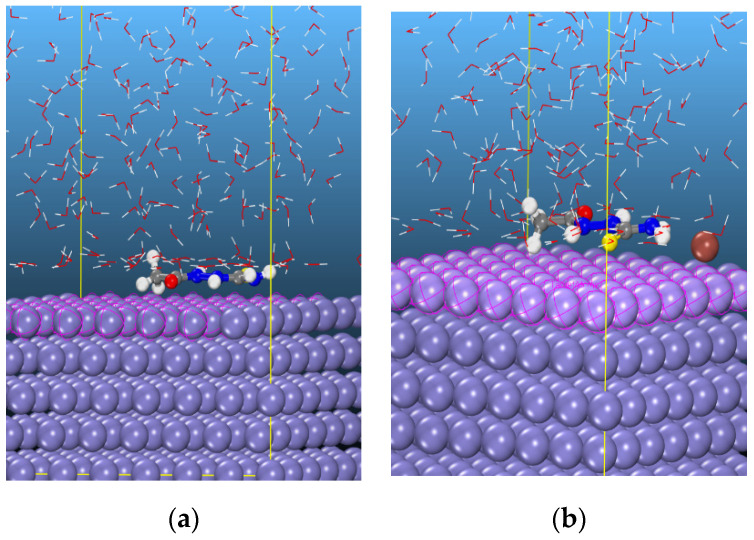
Snapshots of the most stable low energy configurations for the adsorption of (**a**) AST and (**b**) AST + I^−^ on the Fe (110)/400 H_2_O interface obtained using the Monte Carlo simulation.

**Figure 11 materials-13-05013-f011:**
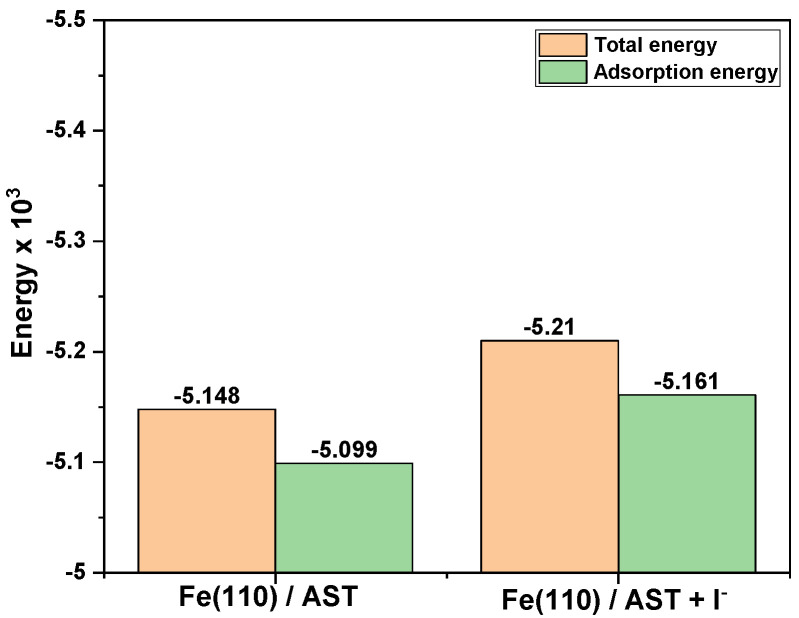
Total energy and adsorption energy calculated by the Mont Carlo simulation for the lowest adsorption configurations of AST and AST + I^−^ systems on Fe (110)/400 H2O Interface at 25°C.

**Table 1 materials-13-05013-t001:** Electrochemical kinetic parameters obtained from LPR measurements for C1018 carbon steel in 1 M HCl (blank) and with various concentrations of AST and AST + KI at 25 °C.

Inhibitor Concentration	*E_corr_* mV	*i_corr_* (µA·cm^−2^)	*R_p_* ohms	CR (mpy)	*η_LPR_* (%)
1 M HCl	−514.2	1580	16.49	138.00	-
250 ppm AST	−505.7	506.3	51.46	44.24	67.94
500 ppm AST	−506.0	481.4	54.12	42.06	69.52
750 ppm AST	−503.9	438.1	59.47	38.27	72.27
250 ppm AST + 5 mM KI	−500.4	290.1	89.82	25.34	81.64
500 ppm AST + 5 mM KI	−498.6	322.5	80.78	28.18	79.58
750 ppm AST + 5 mM KI	−503.6	352.5	73.92	30.79	77.69

**Table 2 materials-13-05013-t002:** Potentiodynamic polarization parameters for C1018 carbon steel in 1 M HCl and various concentrations of AST and AST + KI at 25 °C.

Inhibitor Concentration	*E_corr_* (mV/SCE)	*i_corr_* (mAcm^−2^)	*β_a_* (mV dec^−1^)	−*β_c_* (mV dec^−1^)	CR (mpy)	*η**_PDP_* (%)
1 M HCl	−494.0	561.0	79.30	98.90	49.01	-
250 ppm AST	−496.0	213.0	61.50	98.70	18.64	62.00
500 ppm AST	−495.0	187.0	56.90	100.40	16.37	66.60
750 ppm AST	−492.0	186.0	55.00	107.40	16.27	66.80
250 ppm AST + 5 mM KI	−484.0	116.0	61.40	99.20	10.16	79.30
500 ppm AST+ 5 mM KI	−488.0	136.0	61.30	99.70	11.87	75.80
750 ppm AST+ 5 mM KI	−490.0	170.0	63.00	103.3	14.65	70.10

**Table 3 materials-13-05013-t003:** Electrochemical kinetic parameters obtained from EFM for C1018 carbon steel in 1 M HCl at various concentrations of AST and AST + KI at 25 °C.

Inhibitor Concentration	*i_corr_* (µAcm^−2^)	*β*_a_ (mV dec^−1^)	*β*_c_ (mV dec^−1^)	CR (mpy)	*η_EFM_* (%)	CF(2)	CF(3)
1 M HCl	701.30	67.45	75.63	61.27	-	2.06	2.96
250 ppm AST	378.40	83.70	95.49	33.06	46.04	2.03	2.97
500 ppm AST	369.60	85.65	95.53	32.29	47.30	2.00	3.19
750 ppm AST	350.40	88.50	94.85	30.61	50.04	2.07	2.98
250 ppm AST + 5 mM KI	254.70	87.22	99.58	22.25	63.69	1.91	2.97
500 ppm AST+ 5 mM KI	276.70	88.81	97.67	24.17	60.55	1.70	2.92
750 ppm AST + 5 mM KI	291.00	87.99	101.00	25.42	58.51	1.94	3.08

**Table 4 materials-13-05013-t004:** Electrochemical kinetic parameters obtained from EIS data for C1018 carbon steel in 1 M HCl and various concentrations of AST and AST + KI at 25 °C.

Inhibitor Concentration	$R_{S}$ (Ω·cm^2^)	CPE		$R_{\mathrm{ct}}$ (Ω·cm^2^)	$C_{\mathrm{dl}}$ (*μF*·cm^2^)	χ^2^ × 10^−4^	*η_EIS_* (%)
		Y_0_ (×10^−6^ *S*·*s*^n^·cm^−2^)	m				
1 M HCl	1.48	161.3	0.92	96.77	224.47	2.82	-
250 ppm AST	1.34	120.1	0.91	292.3	162.38	3.07	66.89
500 ppm AST	1.5	115.6	0.9	304.5	152.19	3.41	68.22
750 ppm AST	1.45	105.2	0.99	326.8	108.15	3.44	70.39
250 ppm AST + 5 mM KI	1.66	114.1	0.87	468.9	149.61	2.66	79.36
500 ppm AST+ 5 mM KI	1.46	130.5	0.88	436.9	174.28	2.42	77.85
750 ppm AST+ 5 mM KI	1.54	106.8	0.89	418.6	144.52	2.71	76.88

**Table 5 materials-13-05013-t005:** Comparison of the performance of the proposed corrosion inhibitor with similar inhibitors already reported in the literature.

S/No	Corrosion Inhibitor	Metal	Acid Concentration	Electrochemical Techniques	Inhibition Efficiency	References
1	1-phenyl-4-(4-nitrophenyl)thiosemicarbazide	Maraging steel	1 M HCl	EIS	47.3% @ 0.2 mM	[20]
2	1-ethyl-4(2,4-dinitrophenyl) thiosemicarbazide	C-Steel	2 M HCl	EIS	68.8% @ 16 µM	[23]
3	1,4-diphenylthiosemicarbazide	C-Steel	2 M HCl	EIS	62.6% @ 16 µM	[23]
4	1-ethyl-4-phenylthiosemicarbazide	C-Steel	2 M HCl	EIS	62.6 % @ 16 µM	[23]
5	1-Acetyl-3-thiosemicarbazide	C1018 Steel	1 M HCl	EIS	70.39 @ 750 ppm	Present Study

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
