# Peer review of "Experimental and Theoretical Insights into the Synergistic Effect of Iodide Ions and 1-Acetyl-3-Thiosemicarbazide on the Corrosion Protection of C1018 Carbon Steel in 1 M HCl"

_materials, 2020, doi:10.3390/ma13215013_

Round 1
Reviewer 1 Report
The A. H. Alamri manuscript reports on the experimental and theoretical insights into the synergistic effect of iodide ions and 1-Acetyl-3- 4 thiosemicarbazide on corrosion protection of carbon steel. While the subject of the work appears to be interesting, I recommend a significant reconsideration of the presented data.
1) In the experimental part the procedure of ATS and KI application on the surface is missing.
2) In my opinion the presence of inhibitors on the surface should be confirmed by an analytical method. Also, it would be interesting to confirm that after a long time of immersion in HCl the inhibitor is still present (or not) on the surface.
3) The authors should give a good explanation of why the KI gives the opposite performance of ATS for varied concentrations. Without KI the highest concentration is the best one while in the presence of KI opposite is the case. The explanation given in the paragraphs starting in line 169 is not clear to me.
“ The increase in the inhibition efficiency of ATS with concentration could be attributed to the increase adsorption of ATS molecules on the carbon steel surface with and without KI.”
Is it the correct statement? I concluded from the following part the authors claim opposite observation for KI containing samples.
“The addition of 5mM KI to 250 ppm of ATS gave an optimum protection of 81.69 % to the C1018 carbon steel in 1 M HCl. Beyond this concentration, the adsorption of the molecules reaches a saturation point and ATS molecules begins to desorb from the carbon steel surface even with increased in AST concentration. This subsequently leads to a decrease in the inhibition efficiency as observed in Table 1.”
Firstly, there is no experimental proof of the desorption process for higher ATS concentrations in a presence of KI, so it should not be given as an explanation before proving.
Secondly, in the case of KI absence, 750 ppm of ATS can be well adsorbed on the surface and give the best performance, and additionally, the authors show results of lowering the absorption energy by iodide presence. In such conditions, it is hard to understand why concentrations above 250 ppm in the presence of KI give worst corrosion protection. In my opinion, the reported changes are the key point of this work and it is not well elaborated.
4) What is the molar ratios between KI and varied concentration of ATS? Did the authors consider that differences of the molar rations (constant KI vs 2500, 500, 750 ppm of ATS) can affect the performance? I would suggest conducting experiments keeping the same molar ratio between KI and ATS, especially when there is computational support for the studies.
5) How the cell used for Monte-Carlo simulation was composed of? Did the authors use 1 molecule of ATS and 1 ion of iodide? How does it correspond to experimental systems?
6) ATS or AST? Both versions used in the manuscript and tables.
Due to these issues, the manuscript clearly needs revisions and significant changes, therefore I cannot recommend this paper for publishing in Materials.
Reviewer 2 Report
The manuscript entitled Experimental and theoretical insights into the synergistic effect of iodide ions and 1-Acetyl-3- thiosemicarbazide on the corrosion protection of C1018 carbon steel in 1M HCl, discussing a very interesting topic of corrosion protection by using thiosemicarbazides as corrosion inhibitors , the best practical and economical method to control and avoid aqueous corrosion. Electrochemical techniques are well combined with theoretical studies to further support the experimental result.
However, it also has some drawbacks requiring major revisions as follows:
- Comparison of corrosion results with other best inhibitors/ or industry standards is necessary. Provide a table comparing the performance of the proposed corrosion inhibitors with similar already reported in the literature inhibitors.
- The inhibitor-metal surface interaction mechanism (graph) is needed to be better linked with the data.
- Author declared: “It is clear from the result that ATS + iodide has both the highest negative total and adsorption energies than ATS alone with Fe (110)/400 H2O Interface. This indicates that the presence of iodide has a synergistic effect and enhances the adsorption energy of ATS with the Fe surface. “
Can author emphasizes the beneficial influence of iodide upon inhibitor adsorption?
- Since the acronyms were noted in abstract or first time appears in the paper as: (polarization measurement (LPR), electrochemical frequency modulation (EFM), potentiodynamic polarization (PDP) measurements and electrochemical impedance spectroscopy (EIS)), authors are asked to use only acronyms further. (As example: electrochemical frequency modulation (EFM) and polarization measurement (LPR) were described 5 times).
Round 2
Reviewer 1 Report
The author did not introduce almost any of the suggested changes to the previous version of the manuscript. In my opinion, this work does not provide good quality evidence nor explanations for the observed phenomena. A detailed explanation of my view you can find in the attached word file. For these reasons, I cannot recommend this work for publishing.

Reviewer 2 Report
The manuscript entitled Experimental and theoretical insights into the synergistic effect of iodide ions and 1-Acetyl-3- thiosemicarbazide on the corrosion protection of C1018 carbon steel in 1M HCl was revised according the proposed suggestions, therefore the manuscript can be published in its present form.
Author Response
Manuscript accepted by reviewer 2
Round 3
Reviewer 1 Report
The results presented in the manuscript are correctly described and analyzed, but the same as before I am concerned about the lack of additional experimental prooves. The presented results are interesting and worth presenting for the scientific community, although additional measurements could significantly improve the work quality. Due to the pandemic situation, I understand the difficulty of the author to make additional experiments and I accept the manuscript in the current form. I am looking forward to 2nd part of this project.